# An Exploration of the Effect of the Kleier Model and Carrier-Mediated Theory to Design Phloem-Mobile Pesticides Based on Researching the N-Alkylated Derivatives of Phenazine-1-Carboxylic Acid-Glycine

**DOI:** 10.3390/molecules27154999

**Published:** 2022-08-06

**Authors:** Jinlong Cai, Yongtong Xiong, Xiang Zhu, Jinyu Hu, Yunping Wang, Junkai Li, Jianfeng Wu, Qinglai Wu

**Affiliations:** 1School of Agriculture, Yangtze University, Jingmi Road 88, Jingzhou 434025, China; 2Institute of Pesticides, Department of Plant Protection, School of Agriculture, Yangtze University, Jingmi Road 88, Jingzhou 434025, China; 3State Key Laboratory of Toxicology and Medical Countermeasures and Laboratory of Toxicant Analysis, Institute of Pharmacology and Toxicology, Academy of Military Medical Sciences, 27 Taiping Road, Haidian District, Beijing 100850, China

**Keywords:** Kleier model, Carrier-mediated theory, phloem mobility, phenazine-1-carboxylic acid, synthesis

## Abstract

The Kleier model and Carrier-mediated theory are effective for molecularly designing pesticides with phloem mobility. However, the single Kleier model or Carrier-mediated theory cannot achieve a reliable explanation of the phloem mobility of all exogenous substances. A detailed investigation of the two models and the scope of their applications can provide a more accurate and highly efficient basis for the guidance of the design and development of phloem-mobile pesticides. In the present paper, a strategy using active ingredient-amino acid conjugates as mode compounds is developed based on Carrier-mediated theory. An N-alkylated amino acid is used to improve the pesticide’s physicochemical properties following the Kleier model, thus allowing the conjugates to fall on the predicted and more accessible transportation region of phloem. Moreover, the influence of this movement on phloem is inspected by the Kleier model and Carrier-mediated theory. To verify this strategy, a series of N-alkylated phenazine-1-carboxylic acid-glycine compounds (**PCA-Gly**) were designed and synthesized. The results related to the castor bean seeds (*R. communis* L.) indicated that all the target compounds (**4a**–**4f**) had phloem mobility. The capacity for phloem mobility shows that N-alkylated glycine containing small substituents can significantly improve PCA phloem mobility, such as **4c**(*i*-C_3_H_7_-N) > **4a**(CH_3_-N) ≈ **4b**(C_2_H_5_-N) > **4d** (*t*-C_4_H_9_-N) > **PCA-Gly** > **4e**(C_6_H_5_-N) > **4f**(CH_2_COOH-N), with an oil–water partition coefficient between 1.2~2.5. In particular, compounds **4a**(CH_3_-N), **4b**(C_2_H_5_-N), and **4c**(*i*-C_3_H_7_-N) present better phloem mobility, with the average concentrations in phloem sap of 14.62 μΜ, 13.98 μΜ, and 17.63 μΜ in the first 5 h, which are 8 to 10 times higher than **PCA-Gly** (1.71 μΜ). The results reveal that the Kleier model and Carrier-mediated theory play a guiding role in the design of phloem-mobile pesticides. However, the single Kleier model or Carrier-mediated theory are not entirely accurate. Still, there is a synergism between Carrier-mediated theory and the Kleier model for promoting the phloem transport of exogenous compounds. Therefore, we suggest the introduction of endogenous plant compounds as a promoiety to improve the phloem mobility of pesticides through Carrier-mediated theory. It is necessary to consider the improvement of physicochemical properties according to the Kleier model, which can contribute to a scientific theory for developing phloem-mobile pesticides.

## 1. Introduction

In recent years, pesticides with phloem mobility [1,2] have received considerable attention due to their effective control of vascular pathogens [3] and improved targeting and utilization efficiency, reducing their usage and associated environmental pollution [4]. Since most pesticides do not have phloem mobility, it is necessary to develop strategies to guide the molecular design of pesticides and improve phloem mobility.

A mathematical model to associate phloem mobility with xenobiotic-physicochemical properties (acid dissociation constant and octanol-water partition coefficients, Log Kow and pKa) was established by Kleier et al. [5]. Xenobiotics with a pKa between −0.5~4 and a Log Kow between 3~6 may have phloem mobility. In previous reports, the Kleier model (Figure 1) was verified as a potential method for predicting whether a compound obtained phloem mobility or not [6,7,8,9,10]. For instance, using the Kleier model, N-carboxymethyl-3-cyano-4-(2,3-dichlorophenyl)pyrrole exhibits good phloem mobility [10]. Furthermore, some compounds are absorbed by endogenous carriers in plants, such as glyphosate and paraquat [11,12] (Figure 2). L-type amino acid transporters (LAT1/LAT2) play significant roles in the uptake of glyphosate [11]. Paraquat uptake is involved in polyamine transporter RMV1 and AtPDR11 [12]. Therefore, another approach to converting nonmobile pesticides into phloem-mobile types consists of introducing endogenous plant substances, such as glucose and amino acid peptides, to modify pesticide molecules by click chemistry [13,14,15,16,17,18], which involves a carrier-mediated process. For example, coupling a non-phloem-mobile insecticide with glycine could improve phloem mobility with fipronil-glycine conjugates [15] (Figure 3). Amino acid carriers were found more efficient in translocating phenyl pyrrole conjugates than sugar carriers [16]. Four amino acid transporters, RcLHT6, RcANT15, RcProT2, and RcCAT, may be involved in the glycine–fipronil coupling phloem transport [17]. Thus, the phloem mobility of exogenous substances correlates with their own physicochemical properties and plants’ endogenous carriers.

Phenazine-l-carboxylic acid (PCA) is an antibiotic secreted by *Pseudomonas sp*. M18. [19,20] PCA is a dual-function fungicide capable of the broad-spectrum inhibition of plant pathogens and promoting plant growth [21,22]. It has the characteristics of a broad-spectrum and a high-efficiency. Currently, PCA is registered as a new microbially sourced fungicide for rice in China and has been widely promoted. However, PCA does not have phloem mobility [23,24]. In our previous reports, we have developed a vectorization strategy coupling the PCA to amino acids based on Carrier-mediated theory, which successfully confers phloem mobility to PCA [23,24,25,26,27,28,29]. The PCA was absorbed by the plants in the form of conjugates and then hydrolyzed by amide hydrolase to PCA [29]. (Figure 4). However, the phloem mobility of these couplings should be further improved [23,29]. Meanwhile, some interesting phenomena have been discovered. For example, based on the Kleier prediction model, the conjugates PCA-L-Tryptophan and PCA-L-Tyrosine (Figure 5) should have an excellent diffusion through the membrane, and phloem mobility should be observed. Nevertheless, the experimental results of the phloem sap analysis violate the Kleier model. PCA-L-Tryptophan and PCA-L-Tyrosine were found to have no phloem mobility, but this may be due to the lack of relevant amino acid carriers [24]. Amino acid carriers should more easily recognize **PCA-Gly** to improve phloem mobility, but their phloem mobility was not as satisfactory as expected because they are more hydrophilic with a low diffusion through the membrane [24]. Thus, the single Kleier model or Carrier-mediated theory cannot achieve a reliable explanation of the phloem mobility of all exogenous substances. In the present paper, a novel strategy of combining Carrier-mediated theory and the Kleier model is proposed for the first time to improve compounds’ phloem mobility. On the one hand, based on Carrier-mediated theory, the active ingredient-amino acid conjugate operates as the molecular model; on the other hand, the N-alkylated amino acid conjugate improves the physicochemical properties by following the Kleier model to promote phloem mobility. Then, the capacity of the Kleier model and Carrier-mediated theory to design phloem-mobile pesticides is inspected, which may provide a more accurate and highly efficient basis for guiding the design and development of phloem-mobile pesticides.

To verify this strategy, PCA-glycine conjugate [24] (a compound with phloem mobility synthesized by our research group) was chosen as the molecule model due to the glycine-rich nature of the model plant. Furthermore, a series of the N-alkylated derivatives of **PCA-Gly** were designed and synthesized (Figure 1). Hydrogen linked with a nitrogen atom is substituted by methyl, ethyl, isopropyl, tert-butyl, and phenyl (**4a**–**4f**). Among them, the Glycine fragments guarantee that they can be carried by carriers, and the N-alkylated derivatives will enhance the hydrophilicity via a higher diffusion through the membrane. The phloem mobility of all the coupling compounds was evaluated by ultra-performance liquid chromatography-mass spectrometry (UHPLC-MS) using castor bean seeds (*R. communis* L.) and a castor bean plant model. The relationship between the movement of phloem with the structure of exogenous compounds was discussed by the Kleier model and Carrier-mediated theory.

## 2. Results and Discussion

### 2.1. Synthesis

According to Figure 1, the target compounds were synthesized with four-step reactions. Due to the water sensitivity of intermediate two, the solvents in this study needed to be pretreated to an anhydrous state. Since intermediate two is unstable, it is prepared to react with intermediate one immediately. Compounds **3a**–**3l** were designed to study the structure-activity relationship by performing a series of alkylation steps at the R_1_ position on N and linking the methyl and ethyl groups at the R_2_ position. Compounds **4a**–**4f** were designed to study the phloem mobility by altering the physicochemical properties of the compounds. The structures of the title compounds **3a**–**3l** and **4a**–**4f** were characterized by ^1^H-NMR and a high-resolution mass spectrum (HR-MS) (See Appendix A).

### 2.2. Phloem Mobility in R. communis Seedlings

The phloem mobility of **3a**, **3g**, **4a**–**4f**, PCA, and **PCA-Gly** was evaluated using the *R. communis* seedlings system, which is an ideal biological model that is widely employed to study the phloem mobility of xenobiotics [25,26]. The cotyledons were incubated with each compound of 200 μM for 2 h. The phloem sap was then collected and analyzed using UHPLC-MS.

The detection results for the phloem sap are shown in Table 1. For the cotyledons incubated in the presence of PCA, the fungicide was not detected in the phloem sap even after 5 h. Compounds **3a** and **3g** were not detected, validating our previous experimental conclusions that PCA-amino acid ester conjugates do not have phloem mobility [27]. In contrast, when the cotyledons were incubated with compounds **PCA-Gly** and **4a**–**4f**, these compounds were clearly found in the phloem sap. The test of the **PCA-Gly** shows good reproducibility and indicates the applicability of Carrier-mediated theory.

Notably, compared with **PCA-Gly**, compounds **4a**–**4f** increasingly deviated from the recognizable structure of an amino acid carrier, but four of the compounds (**4a**, **4b**, **4c**, and **4d**) exhibited better phloem mobility. The phloem transport ability was **4c**
*>*
**4a** ≈ **4b**
*>*
**4d** > **PCA-Gly**
*>*
**4e** > **4f** in the castor bean system. Compounds **4a** (CH_3_-N), **4b**(C_2_H_5_-N), and **4c** (*i*-C_3_H_7_-N) had better phloem mobility, with the average concentrations in phloem sap of 14.62 μΜ, 13.98 μΜ, and 17.63 μΜ in the first 5h, which were 8 to 10 times higher than **PCA-Gly** (1.71 μΜ). Compared with our previous studies [23,24,25,26,27,28,29,30,31], compounds **4a**–**4c**’s phloem transportation ability comprised the best class of compounds. The results imperfectly correspond to Carrier-mediated theory, as based on Carrier-mediated theory, **PCA-Gly** should have the best phloem mobility. These findings suggest that the single Carrier-mediated theory cannot achieve a reliable explanation of the phloem mobility of all exogenous substances.

### 2.3. Prediction of Phloem Mobility Using the Kleier Model

The Kleier model is widely used to predict whether xenobiotics have phloem mobility based on their physicochemical properties (log *K*_ow_ and p*K*a) [6,7,8,9,10]. The experimental data fit well with the theoretical predictions for most of the tested xenobiotics. Thus, the physicochemical properties of the compounds **3a**, **3g**, **4a**–**4f**, **PCA-Gly,** and **PCA** are listed in Table 2. Based on their physicochemical properties, we marked the compounds on the predicted phloem mobility in Figure 6.

As shown in Figure 6, compounds **3a**, **3g**, **4a**, **4f**, **PCA-Gly,** and **PCA** were predicted to possibly have certain mobility. Compounds **4b**, **4c**, **4d,** and **4e** were in the moderately mobile compounds’ areas, indicating that these compounds have moderate phloem mobility. This met the design requirements stating that the N-alkylated amino acid conjugate improves the compounds’ physicochemical properties by following the Kleier model, which can lead it to fall on the transportation region that was predicted to be more accessible in phloem. Systemic tests with the *Ricinus communis* seedlings also showed that all the target compounds (**4a**–**4f**) had phloem mobility.

The LogKow is first considered when determining the permeability of exogenous compounds and the capacity for phloem mobility [6,7,8,9,10]. **PCA-Gly** and compounds **4a**–**4e** with the same p*K*a values (3.28–3.31) and LogKow enhanced gradually (1.23–2.61). Simple alkylation did not affect the p*K*a, but significantly improved LogKow. Additionally, the Kleier model (Figure 6) also predicted that the phloem transport ability by compound was **4d** > **4e** > **4c** > **4b** > **4a** > **4f** > **PCA-Gly.** In fact, the phloem transport ability was in the sequence of **4c**
*>*
**4a** ≈ **4b**
*>*
**4d** > **PCA-Gly** > **4e** > **4f**, which contradicts the predictions of the Kleier model. Compared with **PCA-Gly**, phloem sap’s concentration does not increase linearly but in a particular range. The Kleier model does not reasonably explain this phenomenon, but when we consider Carrier-mediated theory, the results fit our hypothesis. The phloem mobility of compounds **4a**–**4c** are consistent with the Kleier model’s theoretical predictions. They have a specific deviation from the identifiable structure of amino acid carriers but can still be effectively combined. The LogKow enhanced gradually (1.23–2.09), enhancing the phloem mobility. Although compounds **4d** and **4e** are more lipophilic than **4a**–**4c**, too much of a deviation in their structures will lead to their reduced recognition or their being unrecognized by amino acid carriers. Therefore, the phloem mobility’s affect is lower than the Kleier model predicted. The synergistic effect began to weaken from compound **4d.** It can be quantified to enable a LogKow between 1.2 and 2.5. Compound **4f** has two free carboxyl groups but fewer detected in the phloem, due to its high hydrophilicity. It was also confirmed that the effect of the octanol–water partition coefficients on exogenous phloem transport is more significant than the acid dissociation constant. Our study verifies the strategy wherein the introduction of plant endogenous compounds as carriers improves the phloem mobility of pesticides by Carrier-mediated theory; simultaneously, it proves the necessity of considering the improvement of the physicochemical properties according to the Kleier model.

### 2.4. Phloem Mobility in Adult Castor Bean Plants

To explore whether compounds **4a**–**4f** could pass through the wax layer, the *R. communis* plant model was used to measure their ability of phloem mobility. Compound **4c**, with the best phloem mobility towards the castor seedlings, was selected as the test compound to screen the experimental conditions. As shown in Table 3, the target compound could move in the phloem without being degraded in detectable amounts during a 24 h test period. This suggests that compound **4c** can pass through the wax layer and accumulate in specific parts of plants. Based on these results, the relationship between the measured values of phloem exudates and the time after applying different concentrations of chemicals is shown in Figure 7. At the concentration of 5 M, the compound **4c** in roots reached the maximum after a 12-h treatment.

According to the phloem mobility of compound **4c** in the castor plants under different conditions, the dosage was 5 M with 12-h treatments to study the phloem mobility of compounds **4a**, **4b**, **4d**, **4e**, **4f**, **PCA-Gly**, and **PCA** under the same conditions (Table 4). All the tested compounds can pass through the wax layer and move in the phloem, except compound **4d** and **PCA**. Among them, the content of compound **4a** reached the maximum in the root more than ten times **PCA-Gly**. Compared with the results of the phloem mobility test of the *R. communis* seedlings, the two results were not wholly consistent. However, the phloem mobility of compounds **4a**, **4b**, and **4c** were still 1–2 orders of magnitude higher than that of **4e** and **4f**. Moreover, the phloem mobility of compound **4e** was also far lower than that of **PCA-Gly**, but it is more lipophilic than glycine. Thus, the structures that deviate too much from the amino acid are not recognized by the carriers, which results in the weakening of the phloem mobility. Compound **4f** and **PCA-Gly** are more hydrophilic and exhibit a small amount of migration in the plants due to the wax barrier.

## 3. Materials and Methods

### 3.1. Chemicals

All reagents and solvents were purchased from commercial suppliers. The melting point was determined by a WRR-Y melting point apparatus (Shanghai Yidian Physical Optical Instrument Co., Ltd., Shanghai, China). Thin-layer chromatography (TLC) was conducted on silica gel plates (GF254) (Qingdao Haiyang Chemical Co., Ltd., Qingdao, China), and spots were visualized on a ZF-I ultraviolet analyzer (Shanghai Gucun Electro-optical Instrument Factory, Shanghai, China). Column chromatography purification was carried out on silica gel (200–300 mesh) (Qingdao Haiyang Chemical Co., Ltd., Qingdao, China). Nuclear magnetic resonance (NMR) spectra were obtained using an AVANCE III HD 400 NMR spectrometer (Bruker Corporation, Basel, Switzerland). Mass spectrographic analysis was conducted on a Thermo Scientific Q Exactive ^TM^ (Thermo Fisher Scientific, Waltham, MA, USA).

### 3.2. Plant Materials

Castor bean seeds (*Ricinus communis* L.) were provided by the Zibo Agricultural Science Research Institute. The castor seedlings were planted as previously reported (Yu et al., 2018). Then, 6-d-old seedlings were selected for the next experiments.

The adult castor bean plants were obtained according to methods described in a previous study [32]. Castor seedlings were grown in nutrient soil in a greenhouse (25–30 °C, natural light) for 3–4 weeks until 3–4 leaves appeared, and cotyledons and primary leaves were removed.

### 3.3. General Synthesis Procedure for Title Compounds ***3a***–***3l*** and ***4a***–***4f***

The synthetic route is described in Figure 1.

#### 3.3.1. General Procedure for Glycine Ester Derivatives 1

As shown in Figure 1, A mixture of R_1_NH_2_ (1 mmol), BrCH_2_COOR_2_ (2 mmol), and K_2_CO_3_ (3 mmol) in DMF (15 mL) was stirred at room temperature for 12 h. Subsequently, 100 mL of water was added to the reaction mixture, and the mixture was extracted three times with 30 mL of ethyl acetate. The organic phase was dried with anhydrous sodium sulfate, filtered, and concentrated in vacuum [33,34].

#### 3.3.2. Synthesis of Phenazine-1-Carbonyl Chloride 2

Phenazine-1-carboxylic acid (2 mmol) was dissolved in 20 mL of anhydrous CH_2_Cl_2_; then, oxalyl chloride (3 mmol) was slowly added. The reaction was stirred at reflux temperature for 8 h. The reaction solution was evaporated under vacuum, and the residue was dissolved in 15 mL anhydrous CH_2_Cl_2_, which was immediately used for the following reaction [28].

#### 3.3.3. General Procedure for PCA-Glycine Ester Derivatives **3a**–**3l**

The glycine ester derivative 1 (2 mmol) was dissolved in CH_2_Cl_2_ at 0 °C, triethylamine (10 mmol) was added, and the reaction was stirred for 15 min. Then, phenazine-1-carbonyl chloride 2 (2 mmol) completely dissolved in 15 mL of anhydrous CH_2_Cl_2_ was added dropwise with respect to the above reaction system. The mixture was stirred at 0 °C for about 6 h until the reaction was complete (monitored by TLC). The reaction solution was washed with a 5% sodium hydrogen carbonate solution and extracted with CH_2_Cl_2_. Then, the organic phase was dried over anhydrous sodium sulfate, filtered, and concentrated in vacuum. Finally, pure target compounds **3a**–**3l** were obtained by column chromatography (PE/EtOAc, *v*/*v* = 4:1) [25].

#### 3.3.4. General Procedure for PCA-Glycine Derivatives **4a**–**4f**

Lithium hydroxide (10 mmol) was added dropwise to a solution of compound **3a** (2 mmol) in water (10 mL) and 1,4-dioxane (10 mL), and the reaction mixture was stirred at room temperature for 5 h until the reaction was complete (monitored by TLC). The 1,4-dioxane and water were removed under vacuum, and the remaining solid was dissolved with a small amount of water. The pH of the aqueous solution was adjusted to 2 with 1 mol/L of HCl. The solid precipitate was then filtered and dried to obtain the pure target compound **4a**. Compounds **4b**–**4f** were also synthesized by this method [24].

### 3.4. Sap Collection from R. communis L. Seedlings

The method of phloem sap collection was the same as that recently described [2,24]. The cotyledons were immersed in a buffered solution containing 200 μmol/L test compounds, and roots were immersed in 500 μmol/L CaCl_2_ solution. After 2 h of incubation, the hypocotyls were cut for phloem exudation. Phloem sap was collected at a 1 h intervals for 5 h. A series of standard solutions (1, 2, 5, 10, and 20 μmol/L) of the test compounds were prepared in methanol for calibration curves. The linear equations of test compounds are shown in Table 5.

### 3.5. Phloem Mobility in Adult Castor Bean Plants

The methodology used for this phase is as follows. Prepare 1 M, 2 M, and 5 M liquid containing the compounds, wrap the upper two true leaves, stem, and matrix soil surface of the castor plant with cling film to avoid contamination of the liquid, and slowly smear the liquid on the lower two true leaves of the castor plant several times with a brush. The amount of liquid medicine used was 1 g, and the castor plants were exposed to natural light in the greenhouse. Repeat the above steps 3 times. Castor root was collected at 3 h, 6 h, 12 h, 18 h, and 24 h and stored at −20 °C for testing.

The pretreatment method of castor samples is as follows. Wash, dry, and section the castor roots. Add 50 mL of methanol with masher crush, add 30 mL of methanol wash segment, transfer to the triangle in the bottle, and seal it in plastic wrap. Conduct an ultrasonic extraction for 30 min, vacuum suction filter, filter residue with an appropriate amount of methanol and ultrasonicate for 10 min, vacuum suction filter again, combine the filtrate, and place the concentration in a rotary dryer until it is near dry to facilitate the following purification.

The purification procedure is as follows. The concentrated extract was transferred to a 250 mL separating funnel with a small amount of dichloromethane; then, 50 mL 10% sodium chloride solution and 5 mL NaOH solution were added. After mixing, 50 mL, 40 mL, and 30 mL dichloromethane was added separately, the extraction was shaken three times, and the lower layer (dichloromethane) was discarded. The pH of the alkaline aqueous phase was adjusted to 3 with 1.6 mL of glacial acetic acid (purity ≥ 99.5%); then, the dichloromethane phase was extracted with 50 mL, 40 mL, and 30 mL dichloromethane three times by shaking, and the dichloromethane phase was collected. After being dehydrated by anhydrous sodium sulfate, the dichloromethane phase was dried by rotation, and the volume was fixed with 5 mL of chromatographic methanol and filtered through a 0.45 μm membrane. A series of standard solutions (0.5, 1, 2, 5, 10, and 20 μmol/L) of test compounds were prepared in methanol for calibration curves. The linear equations of test compounds are shown in Table 6.

### 3.6. Analytical Methods

The phloem sap was diluted with pure water (phloem sap/pure water, *v*/*v* = 1:9), and analyzed by ultra-high performance liquid chromatography mass spectrometer (UHPLC-MS) (Thermo UltiMate 3000 TSQ-Quantis, Waltham, MA, USA). A C18 reversed-phase column (3 um, 100 × 2.1 mm, Thermo Fisher Scientific Co., Ltd., MA, USA) was used for separations at 30 °C. The mobile phase was composed of methanol and water containing 0.1% formic acid with an isocratic elution (methanol/water containing 0.1% formic acid, *v*/*v* = 70:30) at a flow rate of 0.4 mL/min. And the injection volume was 10 μL. The optimized parameters of electrospray ionization in the positive mode were as follows: pos ion spray voltage, 3500 V; sheath gas, 30 Arb; aux gas, 5 Arb; ion transfer tube temp, 350 °C; and vaporizer temp, 400 °C.

## 4. Conclusions

All of the hydrolyzed compounds (**4a**–**4f**) with exposed carboxyl groups exhibited excellent phloem mobility in *R. communis* L. compared to the non-phloem-mobile PCA and PCA-amino acid ester conjugates. The phloem mobility of **4a**–**4c** was significantly enhanced—8 to 10 times higher than **PCA-Gly.** Therefore, the N-alkylation of **PCA-Gly** promotes phloem mobility. Our previous studies have demonstrated that the carboxyl group is an amino acid-carrier binding site [23,24,25,26,27,28,29]. Based on Carrier-mediated theory, N-alkylated amino acid conjugates will increase molecular width and the steric hindrance, resulting in the decrease in the carrier-binding conjugates. Compounds **4a**–**4c** are still within the binding range; thus, their phloem mobility increases with an increasing lipophilicity and exhibit the synergism of Carrier-mediated theory and the Kleier Model. The synergistic effect began to weaken starting with compound **4d.** The *R. communis* L. results indicate that small substituents can significantly improve PCA’s phloem mobility, and this can be quantified to enable a LogKow between 1.2 and 2.5. Compound **4e** is difficult to combine with amino acid carriers due to the considerable steric hindrance of phenyl. Even if the lipophilicity was improved, the movement of the phloem is lower than **PCA-Gly**. Compound **4f** and **PCA-Gly** are more hydrophilic and exhibit a small degree of migration in plants. The experiment involving the phloem mobility in adult castor bean plants showed that most of the tested compounds can pass through the wax layer and move in the phloem. This synergism is similar to that of *Ricinus communis* L. Therefore, we suggest introducing plant endogenous compounds as a promoiety to improve the phloem mobility of pesticides via Carrier-mediated theory. It is necessary to consider the improvement of the physicochemical properties according to the Kleier model. This study verifies that the carrier-mediated theory and Kleier model can play a synergistic role in promoting the phloem transport of exogenous compounds. As far as we know, this theory is the first to combine the Kleier model with the Carrier-mediated theory in the design of phloem-mobile pesticides. We provide an active ingredient-amino acid conjugate structural model, which can also extend to other plant endogenous nutrients, such as glucose, peptides, etc. However, more data are still needed for supplements, which will be further studied. This research and its further iterations will contribute to a scientific theory for developing phloem-mobile pesticides.

## Data Availability

The datasets generated or analyzed during this study are available from the corresponding author on reasonable request.

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
