# Peer review of "An Exploration of the Effect of the Kleier Model and Carrier-Mediated Theory to Design Phloem-Mobile Pesticides Based on Researching the N-Alkylated Derivatives of Phenazine-1-Carboxylic Acid-Glycine"

_molecules, 2022, doi:10.3390/molecules27154999_

Round 1
Reviewer 1 Report
Developing new fungicides with phloem systemicity was believed to be a promising way to control root and vascular pathogens. In this work, the authors try to combine Kleier model with carrier theory for the design of phloem-mobile fungicide. Some compounds showed very good phloem mobility under experimental conditions in this study. In my opinion, the manuscript can be accepted for the publication after being revised. There are some aspects that would help to improve the manuscript's quality.
1. The term “Carrier theory” is not widely used, either giving a detailed definition or citing reference. The carrier means certain plant transporters, the carrier-mediated transport may be involved in the phloem mobility of the design compounds, so “Carrier-mediated theory” is more accurate.
2. Line 35, …endogenous plant compounds as promoiety to…
3. Line 81, …Pseudomonas sp. M18.
4. Lines 216-225, the quantifications of phloem sap and root samples using UPLC-MS were the same? The limits of quantification (LOQs) for each compound should be added in Table 1.
5. Lines 246-251, all the tested compounds in phloem sap were very stable? Some compounds may be metabolized in the phloem, so the tested concentration may be underestimated.
6. Lines 297-306, the discussion was speculative, the carrier-mediated transport of 4a and 4c should be investigated. It is necessary to prove the involvement of carrier-mediated transport in addition to the dissuasion through the membrane.
Reviewer 2 Report
The authors have synthesized a series of N-alkylated derivatives of a previously studied pesticide conjugate (PCA-Gly) and studied their phloem mobility in R. communis. Several compounds with enhanced phloem mobility have been screened out, and both physicochemical properties (Kleier model) and carrier-meditated theory were used to explain the improvement. The manuscript in general presented interesting insights for the design of phloem-mobile of pesticides, and should be considered for publication in Molecules after conducting the following revisions.
1. The manuscript need to be checked thoroughly for grammar and formatting mistakes. Some of them are listed as below:
1) “Ricinus communis” should be italic throughout the manuscript;
2) names of chemicals or substituents: “iso-”(i-) or “tert” (t-) should be italic;
3) misuse of full-width punctuations (e.g. some of the brackets);
4) inappropriate time tenses and letter cases (e.g. line 205 “ml”, line 207 “PH”, title of 2.5, etc.) throughout the manuscript.
2. The authors claimed that the amino-acid carriers play important role in transporting the new compounds into the plant tissues. Other than the analysis of Kleier model, it would be better to provide some experimental evidences for the involvement of active transportation.
3. More recent literatures should be cited to explain the “Carrier theory” in the Introduction (e.g. Wu et al. Pest Manag Sci, 2019, 1507-1516). Consider to use a more propriate designation of the theory.
4. The Conclusion section need to be revised carefully for improved organization and credibility, especially when interpreting the “synergistic effect” and the novelty.
5. Some of the titles of Figures and Tables are too simplistic. More information about the data should be included.
Reviewer 3 Report
1. The study on detecting phloem-mobile pesticides present in the MS is good work, but only theoretically; the MS doesn’t justify the findings.
2. I want to state here that, as the authors have done the work on a theoretical basis, if they can do the microscopic analysis of plant species at different time intervals before harvesting them to know the changes in the phloem tissue during the mobilization of the compounds from roots to the upper part of the plants. This additional work justifies the findings and enhances the work done.
3. There are numerous syntax errors throughout the MS that needs to be addressed.
4. The botanical nomenclature of the plant should be represented in full wherever first cited and the genus name abbreviated thereon.
5. The introduction needs to be more precise.
6. Figure 7: the equation on how the concentrations of the compounds in roots were evaluated needs to be added.
7. Equations for all the experiments need to be incorporated into the MS.
8. The conclusion needs some more authentication of the results obtained during the study.
Reviewer 4 Report
The manuscript “Explore the Effect of Kleier Model and Carrier Theory to Design Phloem-Mobile Pesticides Based on Researching the N-alkylated Derivatives of Phenazine-1-Carboxylic Acid-Glycine” is a continuous investigation of the research group on phloem mobility of PAC. Overall, the manuscript is good and I have recommendations.
1. Line 41: you claim that pesticides were designed to control phloem pathogens: do you mean antibiotics?
2. Please specify exactly how the phloem sap was collected. You mentioned two references and these references cited other references!!! Did you collect from bark tissues or from bark and stem together?
3. The procedure should be in the past tense not as instruction or protocol.
4. Why was castor bean selected?
5. Table 3. The concentrations are in µM. I would like to see them in µM/kg tissue or µg or mg/kg plant tissues. In Table 5 you expressed them in mg/kg.
6. The letters in Table 3 assigned for Duncan’s test should be superscript. The same for table 5.
7. Figure 7, y-axis shows content in root and the caption says compounds in phloem sap !!
8. Table 5 and 6 should be figures as figure 7. You can change the concentrations to µg/kg if you like. This will make higher numbers for the graphs.
9. As I see, the application was made via root. Any justification? What is the application method used in the field? Is it a foliar application?
10. What about the movement of the tested compounds in woody trees? Do they follow the same Kleier model or carrier theory?
Round 2
Reviewer 1 Report
After reading the resubmitted text and the response to comments, I think the manuscript has been carefully and reasonably revised. In my opinion, the manuscript can be accepted for the publication.
